# Glial Fibrillary Acidic Protein: A Biomarker and Drug Target for Alzheimer’s Disease

**DOI:** 10.3390/pharmaceutics14071354

**Published:** 2022-06-26

**Authors:** Akshatha Ganne, Meenakshisundaram Balasubramaniam, W. Sue T. Griffin, Robert J. Shmookler Reis, Srinivas Ayyadevara

**Affiliations:** 1Central Arkansas Veterans Healthcare Service, Little Rock, AR 72205, USA; aganne@uams.edu (A.G.); mbalasubramaniam@uams.edu (M.B.); griffinsuet@uams.edu (W.S.T.G.); 2Department of Geriatrics, University of Arkansas for Medical Sciences, Little Rock, AR 72205, USA; 3BioInformatics Program, University of Arkansas for Medical Sciences, Little Rock, AR 72205, USA; 4Department of Biochemistry & Molecular Biology, University of Arkansas for Medical Sciences, Little Rock, AR 72205, USA

**Keywords:** glial fibrillary acidic protein (GFAP), protein aggregation, protein phosphorylation, post-translational modification (of proteins), Alzheimer’s disease (AD), biomarkers, therapeutic targets

## Abstract

Glial fibrillary acidic protein (GFAP) is an intermediate filament structural protein involved in cytoskeleton assembly and integrity, expressed in high abundance in activated glial cells. GFAP is neuroprotective, as knockout mice are hypersensitive to traumatic brain injury. GFAP in cerebrospinal fluid is a biomarker of Alzheimer’s disease (AD), dementia with Lewy bodies, and frontotemporal dementia (FTD). Here, we present novel evidence that GFAP is markedly overexpressed and differentially phosphorylated in AD hippocampus, especially in AD with the apolipoprotein E [ε4, ε4] genotype, relative to age-matched controls (AMCs). Kinases that phosphorylate GFAP are upregulated in AD relative to AMC. A knockdown of these kinases in SH-SY5Y-APP_Sw_ human neuroblastoma cells reduced amyloid accrual and lowered protein aggregation and associated behavioral traits in *C. elegans* models of polyglutamine aggregation (as observed in Huntington’s disease) and of Alzheimer’s-like amyloid formation. In silico screening of the ChemBridge structural library identified a small molecule, MSR1, with stable and specific binding to GFAP. Both MSR1 exposure and GF AP-specific RNAi knockdown reduce aggregation with remarkably high concordance of aggregate proteins depleted. These data imply that GFAP and its phosphorylation play key roles in neuropathic aggregate accrual and provide valuable new biomarkers, as well as novel therapeutic targets to alleviate, delay, or prevent AD.

## 1. Introduction

Glial fibrillary acidic protein (GFAP) is a type III intermediate filament structural protein predominantly found in astrocytes, which may play causal roles in several age-related neuropathologies [1]. GFAP has been implicated causally in animal models of AD [2] and Alexander’s disease [3,4]. Detection of GFAP in human CSF or serum is a critical biomarker of neuropathology, contributing to diagnoses of Alzheimer’s disease (AD), Parkinson’s disease (PD), dementia with Lewy bodies (DLB), and frontotemporal lobar degeneration (FTLD) [5,6,7]. 

GFAP mutations, favoring aggregation to form Rosenthal fibers, are both causal and diagnostic for chronic gliosis, as found in Alexander’s disease [3], and several single-nucleotide polymorphisms in GFAP are strongly associated with this demyelination disorder [3]. GFAP expression is transcriptionally modulated by multiple growth factors and nuclear hormone receptors [8]. Post-translational modifications (PTMs) previously reported for GFAP include site-specific phosphorylations associated with Alexander’s disease [9], PD [10], and FTLD [11]; acetylations at six lysine residues in amyotrophic lateral sclerosis (ALS) [12]; and citrullination at five arginine residues in AD [13]. Citrullination of GFAP may also contribute to traumatic brain injury [14] and autoimmune disorders [15]. 

GFAP is encoded by a single gene on chromosome 17, expressed as 10 isoforms that differ with regard to splice sites [2,16,17,18]. The principal isoform, GFAP-α (432 amino acids), is highly expressed in central nervous system (CNS) glial cells and neurons, whereas isoforms β (beta), γ (gamma), ε (epsilon), κ (kappa), and ζ (zeta) are expressed in many tissues and cell types beyond CNS neurons and glia [2,16,17].

Neuronal stress, caused by either disease or injury, evokes astrocyte activation as a response, including hypertrophy, proliferation, and increased GFAP expression [19,20,21,22]. Initial post-injury glial activation is an acute response enabling recovery from brain insults [23,24]. Long-term neuronal injury or stress causes chronic neuroinflammation, however, with negative impacts on brain function [25,26]. AD is diagnosed as dementia accompanied or preceded by neuronal pathology comprising amyloid plaques containing Aβ_1–42_ and neurofibrillary tangles derived from tau protein [23,27,28]. We observed elevated GFAP in hippocampi of patients with AD, relative to age-matched controls [29]. 

In the current study, we focus on the roles of GFAP and its PTMs in protein aggregation. We observed hyperphosphorylation and oxidation of aggregated GFAP from AD hippocampus, relative to age-matched control (AMC) GFAP, which appears to lack substantial phosphorylation or oxidation. We pursued AD-specific phosphorylations by RNAi knockdowns of upstream kinases that may target the modified GFAP sites, each of which resulted in a marked reduction in amyloid deposition by human neuroblastoma and glioma cells in vitro. Finally, we screened ~750,000 small molecules from the ChemBridge structural library to identify drug candidates with specific affinity for partially unfolded GFAP. One of these, termed MSR1, was particularly effective in reducing protein aggregation and pathology in a variety of AD models (human cells or *C. elegans*) and served as a lead compound for further pursuit. 

## 2. Methods

### 2.1. C. elegans Strains

The nematode strains used in this study were all obtained from the Caenorhabditis Genetics Center (CGC; Minneapolis, MN). Nematode strains serving as models of AD-like amyloidosis were CL4176 [*smg-1^ts^*; *myo-3p*::*Aβ_1–42_*::*let-851* 3′-UTR; *rol-6(su1006)*], expressing human Aβ_1–42_ in muscle, and CL2355 [*smg-1**^ts^*; *snb-1**::Aβ_1–42_::3′-UTR*(long); *mtl-2::gfp*], expressing human Aβ_1–42_ in all neurons. Strain AM141 expresses a polyglutamine reporter (Q40::YFP) in muscle and thus serves as a model of glutamine-tract aggregation similar to that observed in Huntington’s disease and several other neuropathologies [30]. All strains were propagated at 20 °C on 2% (*w*/*v*) agar plates containing nematode growth medium (NGM), overlaid with *E. coli* strain OP50, unless otherwise noted.

### 2.2. Chemotaxis and Paralysis Assays in Aβ-Transgenic C. elegans Strains CL2355 and CL4176

Transgenic *C. elegans* strains, expressing Aβ_1–42_ in neurons (CL2355) or in muscle (CL4176), were grown at 20 °C with ample *E. coli* (OP50) bacteria, and on day 1 of adult life, worms were lysed to release eggs and thus generate a synchronized cohort. Eggs were then placed on 100 mm NGM-agar Petri dishes seeded with RNAi-expressing bacteria (*E. coli* HT115) targeting a GFAP ortholog or empty-vector control bacteria. Worms at the L3-L4 transition were upshifted to 25.5 °C to induce human Aβ_1–42_ transgene expression and assayed after 48 h. Worms were returned to 20 °C, and chemotaxis [31] and paralysis [32] assays were performed as described [29,33,34,35].

### 2.3. Paralysis Assays in Human Tau-Expressing C. elegans Strain VH255

Transgenic *C. elegans* strain VH255, with pan-neuronal expression of human tau [36], was maintained at 25 °C on agar plates overlaid with a lawn of *E. coli* (OP50) bacteria. Worms were lysed in alkaline hypochlorite solution to obtain unlaid eggs used to initiate a synchronized culture. The eggs were then transferred to 100 mm NGM-agar plates, to which drug candidates MSR1 or MSR2 were added to final concentrations of 1 µM. For *ifp-1* RNAi knockdown, the bacterial lawn consisted of strain HT115 expressing an exonic segment of double-stranded *ifp-1* RNA (see next section). The worms were washed and added to fresh, drug-equilibrated plates every two days. Assays were performed on day-3 adults (i.e., 5.5 days after egg hatching) to assess the fraction of paralyzed worms. 

### 2.4. RNAi in C. elegans

RNA-mediated interference (RNAi) is achieved by feeding bacteria that express double-stranded RNA corresponding to an exonic fragment of the mRNA target [33,37,38,39]. Briefly, worm cultures were synchronized with alkaline hypochlorite lysis (see Section 2.2 and Section 2.3) to release unlaid eggs. Eggs prior to hatch, or late-L4 larvae, were placed onto IPTG-containing NGM plates seeded with bacteria (*E. coli* HT115[DE3]) carrying the empty vector L4440 (pPD129.36) or bacterial clones expressing *ifp-1* (homologous to human *GFAP*)*, let-502* (orthologous to human *ROCK1*), *akt-2* (*AKT2*), *kin-1* (*PKA*), or *grk-1* (*BARK*). Worms on adult day 3 (5.5 days post-hatch) were imaged to assess total aggregate fluorescence (strain AM141) or were assessed for paralysis (VH255) or chemotaxis toward n-butanol (CL2355). 

### 2.5. siRNA Knockdowns and Thioflavin-T Staining of Human Cells

*SH-SY5Y*-APP_Sw_ cells from an exponentially growing culture were trypsinized, rinsed, plated at 10,000 cells per well in 96-well plates, and grown for 16 h at 37 °C in “DMEM + F12” (Life Technologies, Waltham, MA, USA) supplemented with 10% (*v*/*v*) fetal bovine serum (FBS). Cells at ~40% confluence were transfected with short interfering RNA (siRNA) constructs targeting *GFAP* (SAS1_Hs01 00227618), *AKT2* (SAS1_Hs01 00035058), *ROCK1* (SAS1 Hs 00065571), *BARK* (SAS1 Hs 00039321), or *PKA* (SAS1 Hs 00217223), all obtained from Millipore-Sigma (St. Louis, USA). Transfections with the indicated siRNAs were performed using RNAiMax reagent (Life Technologies) according to the manufacturer’s directions. Cells at 48 h post-transfection were fixed in 4% *v*/*v* formaldehyde and stained in a dark container with 0.1% *w*/*v* Thioflavin T. After 4 washes in PBS, cells were covered with Antifade + DAPI (EMD/Millipore-Sigma), and fluorescence was captured in green and blue channels, using a Keyence fluorescence microscope with motorized stage for automated well-by-well imaging, 9 fields per well. Thioflavin-T fluorescence intensity was divided by the number of DAPI^+^ nuclei in each well, yielding ratios of amyloid per cell, summarized as mean ± SD.

### 2.6. MSR1 Treatment of SH-SY5Y-APP_Sw_ Cells

Human neuroblastoma cells were grown as previously described [33,40]. These SH-SY5Y-APP_Sw_ cells, expressing an aggregation-prone “Swedish” double mutant of amyloid precursor protein (APP_Sw_), were maintained in DMEM plus 10% (*v*/*v*) FBS at 37 °C. Cells were suspended in trypsin/EDTA and rinsed in buffer prior to replating or harvesting. Immediately preceding assay, cells were grown for 48 h in the presence of 10 µM MSR1 dissolved in DMSO (0.02% final concentration) or 0.02% DMSO (solvent alone) for control cells. Cells were harvested, total protein was isolated, and insoluble aggregate proteins were purified as described below.

### 2.7. Western-Blotting Analysis of Glial (T98G) Cells for pGFAP and ROCK1

Human glioblastoma cells (T98G) were maintained in Dulbecco’s modified Eagle’s medium (DMEM; Invitrogen/Life Technologies, Grand Island, NY, USA), supplemented to 10% FBS, *v*/*v*. Cells were harvested, and their proteins extracted in lysis buffer (50 mM Tris-HCl, pH 7.5, 150 mM NaCl, 1% *w*/*v* Nonidet P40, 0.1% SDS, 0.5% sodium deoxycholate) and quantified with Bradford reagent (Bio-Rad). Protein aliquots (50 μg) were electrophoresed for 2 h at 100 V on a 4–20% gradient bis-tris acrylamide gel (BioRad Life Science, Hercules, CA, USA) and transferred to nitrocellulose membranes. After pre-incubation with BSA blocker (Pierce/ThermoFisher, Waltham MA, USA), membranes were probed with rabbit antibody to pGFAP or ROCK-1 (Cell Signaling, Danvers MA, USA; 1:100 dilution) overnight at 4 °C. After washes, membranes were incubated for 1 h at room temperature with a secondary antibody, either HRP-conjugated goat anti-rabbit IgG (AbCam, Boston MA, USA; 1:10,000 dilution) or rabbit anti-goat IgG (Rockland Immunochemicals, Gilbertsville, PA, USA), and developed using an ECL chemiluminescence detection kit (Pierce). Data were digitized and analyzed using ImageJ software (NIH).

### 2.8. Isolation of Aggregate Proteins

Cultured human cells were collected, flash frozen in liquid nitrogen, and homogenized at 0 °C in the presence of buffer containing nonionic detergent (1% *v*/*v* NP40, 20 mM Hepes pH 7.4, 300 mM NaCl, 2 mM MgCl_2_, and protease/phosphatase inhibitors [CalBiochem, St. Louis MO, USA]) [11,12,37]. Lysates were centrifuged (5 min, 3000 rpm at 4 °C) to remove debris. After removal of cytosolic proteins (soluble in 1% NP40 nonionic detergent) by centrifugation (18 min, 13,000× *g* at 4 °C), protein pellets were brought to pH 7.4 with 0.1 m HEPES buffer containing 1% *v*/*v* sarkosyl (sodium lauryl sarcosinate) and 5 mM EDTA, and centrifuged for 30 min at 100,000× *g*. Pellet proteins (the sarkosyl-insoluble fraction) were resuspended in Laemmli loading buffer (containing 50 mM dithiothreitol and 2% *v*/*v* SDS, sodium dodecyl sulfate), heated for 5 min at 95 °C to dissolve proteins soluble in this buffer, and separated by electrophoresis on 4–20% polyacrylamide gradient gels containing 1% *v*/*v* SDS. Gels were stained with SYPRO Ruby (ThermoFisher) or Coomassie Blue to visualize protein. Gel lanes were robotically cut into 1 mm slices and digested thoroughly with trypsin. Proteins in each slice were identified using mass spectrometry as described previously [29,35,37,41,42,43].

### 2.9. Modeling and MD Simulation of GFAP Structure

The three-dimensional structure of GFAP was modeled using fold recognition and ab initio structure prediction methods implemented by I-TASSER server-based algorithms. I-TASSER generates 5 different models, of which the lowest-energy conformer was chosen for further processing. For molecular dynamic (MD) simulations, we used the protein-preparation wizard within the Schrödinger Desmond simulation suite to prepare modeled structures. Physiological conditions were approximated during simulations by creating an orthorhombic simulation box filled with simple point charge (SPC) water, neutralization of locally charged sites with appropriate counterions (Na^+^, Cl^−^), and further addition of 0.15 M NaCl to achieve a physiological isotonic state. For equilibration, temperature and pressure were held at 300 °K and 1.1023 bar, respectively. The random sampling seed input was changed for each run, and each 200 to 500 ns simulation was repeated at least three times. Simulations with phosphorylations were incorporated using the Maestro “Mutate residue” plug-in, in which specified residues were converted to their phosphorylated form. Trajectories were visualized with VMD and analyzed using BIOVIA Discovery Studio (Dassault Systemes, Waltham, MA, USA).

### 2.10. Virtual Screening of a Target Protein against Molecular Structure Libraries

High-throughput virtual screening of the ChemBridge molecular structure library for docking to GFAP was initially conducted using the Glide module of Schrödinger Suite. ChemBridge structures were retrieved in 2D format and prepared using LigPrep wizard (Schrödinger Suite). A 3-stage strategy was employed to improve the efficiency of virtual drug screening: (*i.*) the entire library of ~750,000 molecular structures was virtually docked to the GFAP protein in Glide’s high-throughput mode; (*ii*.) the top 1% of structures from high-throughput screening were then docked again to GFAP in Glide’s standard precision mode; and (*iii*.) for the top 1% of structures from standard precision docking, binding free energies were predicted under MM-GBSA conditions using the Schrödinger Suite Prime module. Protein–ligand complexes with highest avidity (lowest ΔG_binding_) were simulated using the Schrödinger Desmond module to assess their stability over time.

### 2.11. Statistical Analyses

For replicated assays of protein aggregation, chemotaxis, and paralysis, differences between control and experimental groups were assessed for significance with the Fisher–Behrens heteroscedastic *t*-test (appropriate for samples of unequal or unknown variance), treating each experiment as a single point. Within experiments, differences in proportions (fractional paralysis or chemotaxis) were evaluated with chi-squared or Fisher exact tests, as appropriate, based on sample counts.

## 3. Results

### 3.1. Glial Fibrillary Acidic Protein Is Enriched, Hyperphosphorylated, and Oxidized in Aggregates Formed in Alzheimer’s Hippocampus

We previously identified the protein constituents of sarkosyl-insoluble aggregate fractions from AD hippocampi [29]. Glial fibrillary acidic protein (GFAP), a largely unstructured protein, is enriched 2- to 2.5-fold in three subclasses of detergent-insoluble aggregates from AD hippocampus relative to age-matched control (AMC) hippocampus (Figure 1a). Although GFAP in control aggregates has no prevalent post-translational modifications (PTMs), GFAP in AD aggregates is phosphorylated at three to five serine or threonine residues (Figure 1b). Remarkably, the GFAP phosphorylation signature differed reproducibly between individuals carrying ApoE alleles ε3,ε3 or ε4,ε4 (abbreviated to “3,3” and “4,4”, respectively). Western blots of AD(3,3) and AD(4,4) samples confirmed significant enrichment of hyperphosphorylated GFAP (hP-GFAP) in AD tissue relative to AMC (Figure 1c); each genotype group differed from AMC(3,3) control samples at *p* < 0.0001.

We also used PEAKS software (PTM module) to screen several other PTMs, but none were useful in distinguishing AD from AMC. In all groups, methylated arginine and lysine were observed at R88 and K95, whereas K107 was dimethylated, all >90%. Deamidation was not observed in >10% of spectral counts for any GFAP peptide, and pyroglutamine never exceeded 25% (data not shown).

### 3.2. Molecular Dynamic Simulations Predict GFAP Unfolding and Identify a Druggable Pocket

Molecular dynamic simulations of the hP-GFAP structures observed in AD aggregates (rendered by phosphomimetic substitutions) predict a more malleable GFAP structure in ApoE(3,3) individuals but relatively greater structural rigidity in AD(4,4) compared to unphosphorylated GFAP, as seen in AMC aggregates (Figure 2). Since GFAP is a largely disordered protein, its full-length structure has not been experimentally determined. We therefore predicted its three-dimensional structure using fold recognition and ab initio procedures in I-TASSER [44]. The resulting hypothetical structure comprises helices and loops (Figure 2a), forming a small pocket or cavity near the inner groove of the protein (yellow region).

In view of its disordered nature, the predicted protein structure is expected to be unstable and to unfold spontaneously, altering the orientation and drug accessibility of the pocket. To explore the protein unfolding trajectory, we simulated the predicted structure of fully solvated GFAP for 0.5 μs (500 ns). The volume of the druggable pocket (Figure 2b), and several measures of atomic positional variation (see below), provide useful descriptors of structural change. These analyses support the anticipated unfolding of the initial GFAP structure, wherein the druggable pocket (shown in yellow, Figure 2a,c) expands during the course of the simulation. We selected an intermediate, metastable structure (at 200 ns of simulation; Figure 2c) to screen for small-molecule binding.

The observed differential phosphorylation of GFAP in AD aggregates (Figure 1) may be expected to impact GFAP structural dynamics. To evaluate this possibility, we computationally “mutated” the observed phosphorylated sites to glutamic acid (phosphomimetic substitutions) and simulated the resulting structures for 0.5 μs. The results support the expectation that GFAP hyperphosphorylation in the AD brain is likely to alter its structural stability. Root mean square deviation (RMSD) of the GFAP atomic coordinates over this time interval is consistently lower for the unmodified molecule, “AMC(3,3) GFAP”, than for “AD(3,3) GFAP”, a phosphomimetic structure similar to that observed in AD(3,3) aggregates (Figure 2d). “AD(4,4) GFAP” (a phosphomimic of GFAP observed in AD(4,4) aggregates) is initially a little more variable than AMC(3,3) GFAP but achieves a relatively stable structure, which is maintained from ~235 ns onward (Figure 2d).

We also plotted the average root mean square fluctuation (RMSF) of individual residues over time, which indicates moderate to high positional fluctuation across the GFAP molecule, for both AMC(3,3) and AD(3,3) structures (Figure 2e), in contrast to the relative rigidity of the AD(4,4) phosphomimetic structure. Together, these data (Figure 2d,e) support the prediction that AD-associated differential phosphorylations can modify GFAP structure, somewhat unstably in ApoE(3,3), but creating a relatively invariant conformation in ApoE(4,4).

### 3.3. Identification of Potential Kinases Mediating GFAP Phosphorylation

Using phosphorylation prediction programs, NetPhos (http://www.cbs.dtu.dk/services/NetPhos, accessed 18 October 2019) and GPS (http://gps.biocuckoo.org/online.php, accessed 18 October 2019), we predicted the upstream kinases for each putative GFAP target residue (Appendix A and Figure 3a). All of the implicated kinases (AKT2, ROCK1, BARK/GRK, and PKA), predicted to phosphorylate GFAP at the modified sites, had been previously implicated in AD pathogenesis or progression [45,46,47,48,49,50,51,52,53]. We therefore tested the effects of individual kinase knockdowns on protein aggregation in human neuroblastoma cells (SH-SY5Y-APP_Sw_; Figure 3b,c) and human glioblastoma cells (T98G; Figure 3d,e), with or without siRNA-mediated knockdown. KD of each kinase gene reduced aggregates in SH-SY5Y-APP_Sw_ cells by 60–70%, similar to (or exceeding) the effect of GFAP siRNA (Figure 3c). In T98G cells, only *AKT2* siRNA relieved aggregation as effectively as GFAP siRNA (by ~50%), but the other kinase knockdowns reduced aggregate fluorescence by 23–28% (Figure 3e).

We also compared the impact of kinase-targeted knockdowns in *C. elegans* models of protein aggregation, using RNAi constructs to silence the closest nematode orthologs of these human kinases. We used aggregation models that simulate Huntington’s disease via polyglutamine array aggregation (strain AM141) and Alzheimer’s disease using neuronal expression of Aβ_1–42_ peptide to form amyloid (CL2355) or muscle expression of human tau to form toxic aggregates that lead to paralysis (VH255). In the Huntington model, total aggregate intensity per worm (Appendix A), i.e., the product of aggregate count per worm and mean YFP fluorescence per aggregate, was reduced 50–60% by knockdown of *C. elegans* genes orthologous to *BARK/GRK* or *ROCK1* (each *p* < 0.00005), similar to the effect of RNAi against *ifp-1* (an intermediate filament protein with extensive homology to human *GFAP*). In a *C. elegans* model of AD-like neuronal amyloidosis (CL2355), chemotaxis declined after neuronal induction of an Aβ transgene, causing fewer worms to migrate toward n-butanol (chemo-attractant). RNAi knockdowns of *ROCK1, AKT2*, or *BARK/GRK* orthologs (*let-502*, *akt-2*, and *grk-2*, respectively) rescued the chemotaxis defect at least as well as KD of *ifp-1,* largely homologous to human *GFAP* (Appendix A). These *C. elegans* results are similar to those observed in the human glioblastoma cell line, T98G, in which siRNA KD of *ROCK1*, *AKT2*, or *PKA* reduced total aggregate protein by 50–60% (Appendix A).

### 3.4. ROCK1 Is Increased in ApoE-Expressing Glioblastoma Cells

The Rho-associated protein kinase 1 (ROCK1) protein level is elevated in mild cognitive impairment and AD, and its reduction by hemizygous knockout blunted the high amyloid levels seen in a mouse model of AD [45]. Since we observed a significant benefit from reducing ROCK1 levels both in human cells in culture and in intact *C. elegans* aggregation models, we measured its levels in T98G glioblastoma cells that overexpress either the *APOE3* or *APOE4* allele. ROCK1 protein levels were at least 6-fold higher in human glial cells that overexpress the *APOE4* allele than in the same cells expressing an *APOE3* transgene (*p* < 0.0001; Figure 4a,b), potentially contributing to the additional GFAP phosphorylation sites observed in AD aggregates from ApoE(4,4) vs. ApoE(3,3) individuals (Figure 1b,c).

### 3.5. Computational Screening Identifies Novel Small Molecules Predicted to Bind Stably to GFAP

To identify novel GFAP-specific inhibitors, we screened structures from the ChemBridge library, comprising ~750,000 small molecules, using in silico docking simulations. For target-based docking, we chose the predicted druggable pocket in the transitional GFAP structure used previously (Figure 2c, at 200 ns). To improve the predictive throughput, we conducted computational screening in three stages, increasing docking stringencies at each stage (see Methods), within the Schrödinger Glide docking module [54]. We first performed virtual docking of the entire ChemBridge library in high-throughput mode, followed by redocking of the top 1% of lead molecules in standard precision mode. The top 1% of molecules emerging from stage 2 (74 structures) were analyzed for implicit-solvent-based free energy of interaction, using Schrödinger’s MM-GBSA module [54]. The molecules predicted to have the lowest ΔG_binding_ to GFAP were pursued in vivo for their impact on protein aggregation. This three-stage procedure for simulated docking produced a set of molecules predicted to bind avidly and stably to the GFAP target pocket. The three best candidates (labeled MSR1, MSR2, and MSR3) were predicted to have ΔG_binding_ surpassing – 46 kcal/mol and to fit well within the modeled GFAP pocket (Figure 2c). Counter-screening for binding to tubulin predicted negligible affinity of these drugs to α or β tubulins, or to oligomers of α and β tubulin, although most other top-ranked drugs were predicted to bind tubulin as avidly as they bound GFAP (data not shown).

### 3.6. A Top-Ranked Drug Suppresses In Vitro and In Vivo Aggregation as Effectively as Knockdown of GFAP and Suppresses the Same Co-Aggregate Constituents

Three lead compounds with the lowest predicted ΔG_binding_ values (Figure 5a) were pursued for experimental validation in a human cell culture model of neurodegenerative amyloidosis and in *C. elegans* whole-animal models of AD-like aggregation. MSR3 was found in initial testing to be cytotoxic to neuroblastoma cells and to delay *C. elegans* development at all doses tested (data not shown) and was therefore not pursued. MSR1 and MSR2 were initially tested for in vivo efficacy in SH-SY5Y-APP_Sw_ neuroblastoma cells; aggregation in this model was especially well suppressed by MSR1 (Figure 5b,c). In all assays, MSR1 was superior or similar in efficacy to MSR2. Figure 5b shows SH-SY5Y-APP_Sw_ neuroblastoma cells stained for amyloid with thioflavin T after exposure to vehicle (control) or MSR1; in multiple experiments, thioflavin fluorescence declined approximately 2-fold in MSR1-treated cells. Total sarkosyl-insoluble aggregate proteins were isolated and separated on acrylamide-SDS gels, and protein was then stained with SYPRO Ruby. Gel lanes (Figure 5d, left to right) show vehicle-treated control cells, cells treated for 48 h with short interfering RNA (siRNA) targeting *GFAP*, or cells treated for 48 h with MSR1 or MRS2, drugs predicted to bind stably and selectively to GFAP. *GFAP* siRNA suppressed aggregate protein by 65–80%, while MSR1 provided 60–75% suppression, but MSR2 did not significantly reduce the amount of aggregate protein.

Proteomic identification of proteins in sarkosyl-insoluble aggregates revealed that proteins completely excluded from such aggregates by MSR1 treatment coincided remarkably well (87% concordance) with those eliminated by siRNA knockdown of GFAP (Figure 5e), whereas four proteins were found to be substantially upregulated by both treatments. The excluded group (which includes synapsin-1, plectin, dynactin-1, MAP1A, MAP2, and α-tubulin) has a correlation coefficient of 0.77 between the effects of MRS1 and GFAP siRNA (*p* < 3 × 10^−^^280^; Figure 5f). This observation of highly concordant and correlated effects of MSR1 and *GFAP* RNAi on aggregate composition provides compelling evidence that GFAP is the chief functional target by which MSR1 lowers aggregation.

Drugs MSR1 and MSR2 were also tested in several *C. elegans* models of human neurodegenerative diseases. In a tauopathy model strain (VH255) expressing normal human tau in *C. elegans* muscle, tau aggregation caused paralysis that was alleviated to a similar extent by treatment with 1 µM MSR1 or siRNA against *ifp-1,* the closest nematode homolog of *GFAP* (Appendix A). MSR1 also effected significant rescue of chemotaxis in a *C. elegans* model of neuronal amyloidosis (CL2355), in which migration toward a chemo-attractant was impaired by pan-neuronal expression of human Aβ_1–42_. The addition of 0.1 µM MSR1 restored chemotaxis to ~90% (Appendix A), the same level as in wild-type or uninduced worms of the same age (not shown). Fluorescent muscle aggregates accumulate with age in a strain expressing Q40::YFP in muscle, mimicking the polyglutamine array expansion threshold for huntingtin protein, i.e., the array length sufficient to elicit symptoms of Huntington’s disease in humans and paralysis in nematodes. Appendix A shows aggregate intensity per worm at 5 days of age post-hatch, reduced ~50% by 10 µM MSR1, vs. 35% at 0.1 µM (each treatment differing from vehicle-only controls at *p* < 0.005).

## 4. Discussion

Aging is the most influential non-genetic risk factor for a variety of dementias and also for many other adult onset diseases that impose significant burdens on aging adults and on our healthcare system [55]. Neurodegenerative diseases, such as AD, Parkinson’s, and amyotrophic lateral sclerosis (ALS), as well as conditions as diverse as hypertension [35], sarcopenia [37], and even several adult cancers [56], all show accrual of distinctive aggregate foci featuring disease-specific “diagnostic” proteins. We identified many proteins within immunopurified aggregate subsets by proteomics [29] and defined protein–protein interfaces by intra-aggregate cross-linking and recovery of linked peptides [57].

GFAP, one of the numerous proteins enriched in AD aggregates relative to age-matched controls [29], now joins a small set of neuropathology-associated proteins that display disease-specific hyperphosphorylation. These include tau (AD, PD, ALS), Aβ_1–42_ (AD), TDP43 (ALS), and α-synuclein (PD) [29,58,59,60,61,62,63]. We identified three GFAP serines that are highly phosphorylated in aggregates isolated from hippocampi of AD(3,3) individuals and additional serines and a threonine near the N- and C-termini for which phosphorylation was observed only in AD(4,4) aggregates (Figure 1c). We examined only *APOE3* or *APOE4* homozygous tissue, but it is reasonable to expect an intermediate outcome for *APOE3/E4* heterozygotes. We also observed three methionine oxidation sites unique to AD-aggregate GFAP (Figure 1c), which may reflect either a more oxidative cell environment or greater misfolding of GFAP molecules in AD hippocampus.

The sites we observed for GFAP phosphorylation in AD are consistent with the known targets of several kinases previously implicated in AD pathogenesis. These include AKT2 (one of two mammalian AKT paralogs), Rho-associated Kinase 1 (ROCK1), G-protein-coupled Receptor Kinase 2 (BARK/GRK2), and Protein Kinase A (PKA). Altered insulin signaling has been implicated in AD, and AKT is a key downstream effector of the kinase cascade that conveys insulin/insulin-like signaling [64,65,66]. GRK2, also known as BARK, was also shown to play a role in the development of cardiovascular disease. PKA, a cAMP-dependent kinase, is involved in multiple signaling pathways, contributes to tau hyperphosphorylation, and has been implicated in progression of several neurodegenerative disorders, including AD, PD, and HD [49,67,68,69,70]. PKA was also shown to play roles in diabetes [70] and anxiety-related behavior [71].

Knockdowns of several kinases that could participate in the observed GFAP hyperphosphorylations decreased protein aggregation and associated physiological declines in *C. elegans* and in human neuroblastoma cells expressing amyloid precursor protein. We note that several of these same kinases have putative target sites in other AD-associated proteins, such as microtubule-associated protein tau, which are expected to further amplify their impact. Multiple AD-promoting targets of these kinases are suggested by results shown in Figure 3c,e, in which some kinase knockdowns provide more effective rescue from AD-like traits than knockdown of GFAP itself. We note, however, that the neuronal efficacy of siRNA knockdowns was not monitored in these experiments and is typically lower in neurons than in other target cells. While the noted comparisons may thus be misleading, a reduced siRNA efficacy in neurons would lead to *underestimation* of the target’s impact.

Based on our data, GFAP provides a novel target to relieve aggregate burden in AD and possibly other aggregation-associated diseases. We therefore screened for small molecules that specifically target GFAP, beginning with the three-dimensional structure of GFAP predicted via powerful fold recognition and ab initio procedures [44]. The initial lowest-ΔG structure included a small druggable pocket (Figure 2a). Molecular dynamic simulations of the GFAP structure over time revealed a transition to a metastable state in which the binding cavity was expanded ~3-fold over its initial volume. The molecular structure of GFAP at 200 ns (Figure 2c) was selected as the target for drug screening. Several descriptors (RMSD, RMSF, and pocket volume) were used to monitor GFAP structural change during the simulation.

Proteomics of brain aggregates indicated strikingly differential phosphorylation of GFAP from AD tissue [29]. Although phosphomimetic substitutions are not the perfect mimics of actual protein phosphorylation, the computer simulations of the predicted AD(3,3) and AD(4,4) structures, incorporating pseudo-phosphorylated sites, as observed, were fully consistent with the hypothesis that phosphorylation status can alter GFAP structural dynamics. Simulation data do not permit us to infer the extent to which phosphorylation of any individual kinase target is responsible for destabilizing the GFAP structure or might alternatively favor accessibility of subsequently modified kinase sites.

We note that our strategy for computational screening of large drug structure libraries has several novel features. These include a three-tier docking screen with progressively increasing stringency and the elimination of drugs predicted to display off-target affinity toward tubulin chains, a common property we observed for most GFAP-binding drugs. We tested the top three candidates emerging from three successive screens for GFAP binding, plus a counter-screen to eliminate tubulin-binding drugs, for efficacy in a variety of aggregation model systems. Of these, MSR1 displayed an efficacy close to that of GFAP knockdown (using RNAi to GFAP or its closest nematode homolog) in each assay, and proteomic analysis of aggregates revealed a near-identical set of proteins depleted or eliminated from aggregates.

## 5. Conclusions

The present study is the first to demonstrate the impact of GFAP and its predicted upstream kinases on protein aggregation in human-cell and *C. elegans* models of neuropathic aggregation and the first successful in silico screen for a GFAP-specific drug, coupled to in vivo demonstrations of its anti-aggregative efficacy. Because the protective effects of MSR1 in nematode models of neurodegenerative pathology are paralleled by similarly positive results in cultured human neuroblastoma cells, GFAP and AD-associated kinases hold promise as novel targets for drug interventions to ameliorate AD-like neuropathies.

## Figures and Tables

**Figure 1 pharmaceutics-14-01354-f001:**
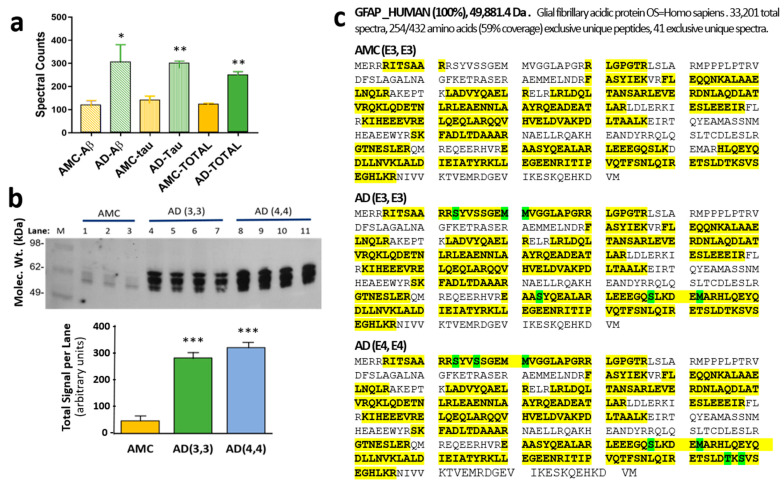
**GFAP is more abundant and more modified in AD aggregates than in AMC.** (**a**) Spectral counts for GFAP in sarkosyl-insoluble aggregates isolated by immuno-pulldown (IP) of Aβ or tau, or in total aggregates without IP, from human AD vs. age-matched-control (AMC) hippocampus. AD differs from AMC according to heteroscedastic *t*-tests: * *p* < 0.05; ** *p* < 0.005. (**b**) Western blot of phosphorylated GFAP from AMC(3,3) vs. AD(3,3) or AD(4,4) hippocampal aggregates, detected with antibody to phospho-GFAP (Ser13; ThermoFisher). *** Each AD group differs from AMC according to two-tailed heteroscedastic *t*-test at *p* < 0.0001. (**c**) Differential post-translational modifications (PTMs: phosphorylation of **S**er or **T**hr residues, or oxidations of **M**et) observed in GFAP isolated from AMC [ApoE(3,3)], AD [ApoE(3,3)], or AD [ApoE(4,4)] hippocampus, as indicated. Peptide coverage is indicated by yellow highlighting, including PTMs indicated by green highlighting.

**Figure 2 pharmaceutics-14-01354-f002:**
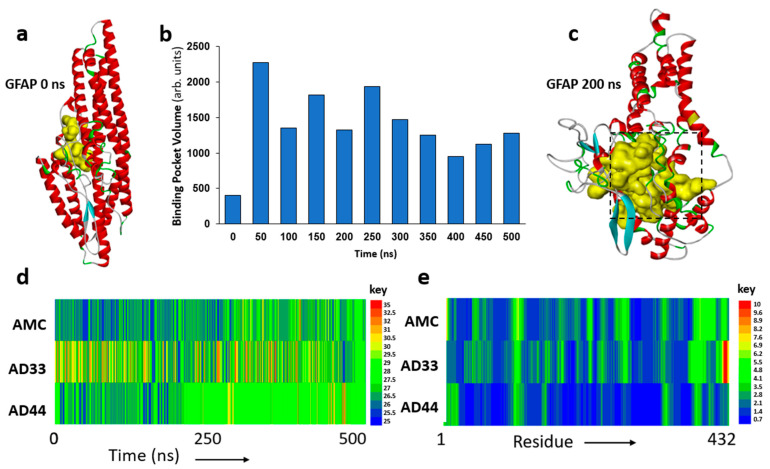
**Molecular dynamic analyses of GFAP structure.** (**a**) Initial modeled structure of GFAP. Yellow internal cavity is the predicted drug-binding pocket. (**b**) Predicted binding pocket volume for GFAP at 50 ns intervals over a 500 ns span. (**c**) Predicted cavity for ligand binding (yellow) at 200 ns. (**d**) Time course of root mean squared deviation (RMSD) for GFAP structure, comparing 500 ns in silico simulations of AMC (unmodified) GFAP to GFAP with phosphomimetic substitutions to mimic AD(3,3) and AD(4,4). (**e**) Distribution across GFAP (432 residues) of root mean squared fluctuation (RMSF) to illustrate positional variability by residue. (**d**,**e**) Keys to the right of panels **d** and **e** display the color codes for RMSD and RMSF values, respectively.

**Figure 3 pharmaceutics-14-01354-f003:**
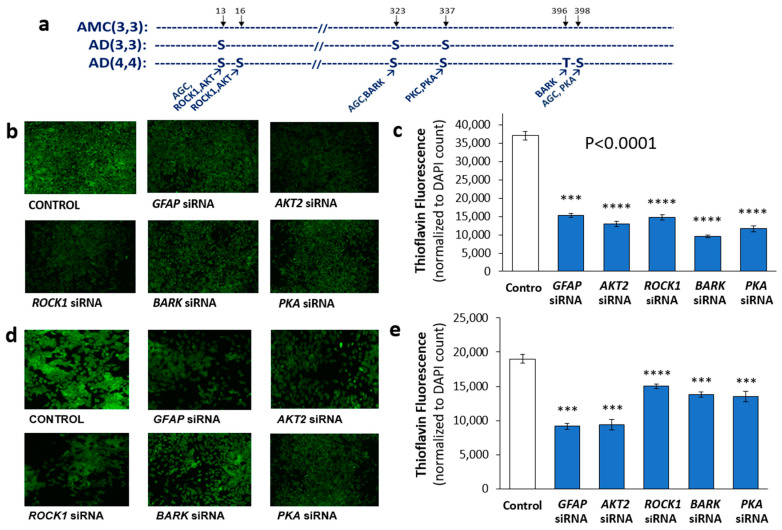
**Effects on aggregation of RNAi knockdowns targeting GFAP or its putative kinases.** (**a**) Observed GFAP phosphorylations in human hippocampal aggregates and their putative kinases. (**b**) Fluorescence images of Thioflavin T-stained SH-SY5Y-APP_Sw_ cells, after liposome-mediated transfection with siRNA constructs targeting GFAP or its candidate kinases. (**c**) Histogram showing means ± SEM for Thioflavin T staining of aggregates as in (**b**). (**d**) T98G cells stained with Thioflavin T after transfection by siRNA constructs as shown. (**e**) Histogram of means ± SEM for Thioflavin T staining of aggregates as in panel (**d**). (**c**,**e**) Significance of differences between treated groups and controls according to heteroscedastic, two-tailed *t*-tests: *** *p* ≤ 0.0005; **** *p* ≤ 0.00005. Experiments were replicated 3–4 times, with consistent results.

**Figure 4 pharmaceutics-14-01354-f004:**
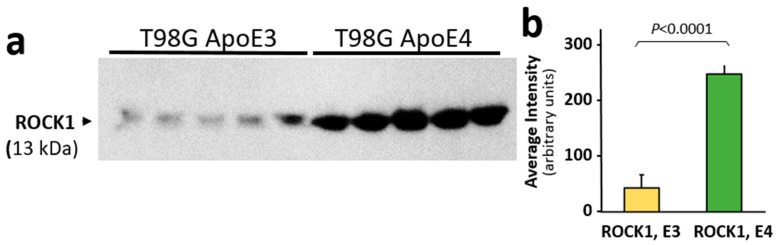
ROCK1 protein levels are higher in T98G glioblastoma cells overexpressing an *ApoE4* transgene than in T98G cells overexpressing *ApoE3.* (**a**) Western blot probed with antibody to ROCK1 protein. (**b**) Mean ± SEM of Western blot band intensity for independent T98G cell cultures (each *N* = 5). The difference between ROCK1 (E3) and ROCK1 (E4) is significant according to heteroscedastic, two-tailed *t*-test at *p* ≤ 0.0001.

**Figure 5 pharmaceutics-14-01354-f005:**
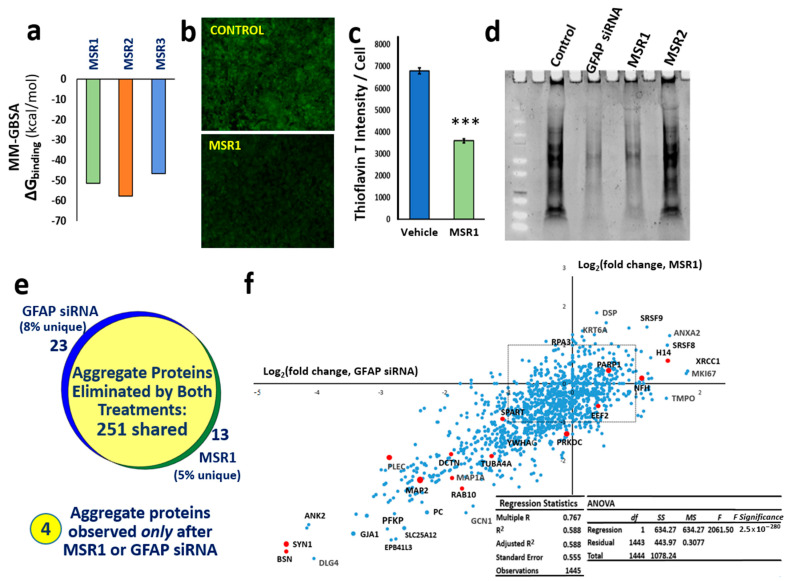
**The drug MSR1 specifically binds GFAP and blocks its role in aggregation**. (**a**) Histogram shows stability (Gibbs free energy of binding) of the top three drugs from in silico screening for solvated MM-GBSA docking to GFAP. The ChemBridge structural drug library was screened in three stages of progressively increased stringency, followed by counter-screening to eliminate drugs with tubulin affinity. (**b**) SH-SY5Y-APP_Sw_ cells were stained with Thioflavin T (green fluorescence) and counter-stained with DAPI (not shown). (**c**) Histogram showing 2-fold reduction in amyloid per cell (thioflavin-T fluorescence divided by the count of DAPI^+^ nuclei per field); *** *p* ≤ 0.0005. (**d**) Total sarkosyl-insoluble aggregate protein, electrophoresed and stained with SYPRO-Ruby after isolation from SY5Y-APP_Sw_ cells. (**e**) The set of proteins totally removed from aggregates by siRNA knockdown of GFAP is nearly identical to the set eliminated by MSR1. The Venn diagram shows proteomic overlap of 251 proteins identified in sarkosyl-insoluble aggregates from untreated SY5Y-APP_Sw_ cells (≥7 hits) but not detected 48 h after transfection with GFAP siRNA or after treatment with 1 µM MSR1. Conversely, four proteins absent from untreated cell aggregates were identified in both treated cell groups. (**f**) Linear regression of log_2_ (fold change) of aggregate protein abundance after GFAP siRNA treatment (X-axis) vs. MSR1 treatment (Y-axis). Selected proteins are labeled, including many found by aggregate cross-linking to be immediate neighbors of GFAP (red dots). Dots within the dashed rectangle were shifted <2-fold by either treatment. *R* = 0.77 for the regression; *F*-test significance was *p* < 3 × 10^−280^.

## Data Availability

Detailed data will be provided upon request, unless prevented by intellectual property considerations.

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
