# Peer review of "Glial Fibrillary Acidic Protein: A Biomarker and Drug Target for Alzheimer’s Disease"

_pharmaceutics, 2022, doi:10.3390/pharmaceutics14071354_

Round 1
Reviewer 1 Report
This manuscript presents results on glial fibrillary acidic protein (GFAP) and predicted kinases on protein aggregation in human cells and C. elegans models. This is a comprehensive and well executed study. This study demonstrates the impact of GFAP on protein aggregation as well as a first successful in silicio screen for GFAP-specific drug model. The study included in vivo illustration of the model drug’s anti-aggregation efficacy. The value of this study relates to the link between GFAP and diseases like Alzheimer’s disease.
The manuscript is a very well written manuscript and can be considered for publication.
Author Response
Thank you for your positive comments!
Reviewer 2 Report
In this work, the authors focused on the roles of GFAP and its PTM in protein aggregation. They observed hyperphosphorylation and oxidation of aggregated GFAP from the AD hippocampus, relative to age-controlled GFAP (AMC), which did not appear to have significant phosphorylation or oxidation.
The authors followed the specific phosphorylation of AD by knockdown of RNAi upstream kinases that could target modified GFAP sites, each of which resulted in a significant reduction in amyloid deposition by human neuroblastoma and glioblastoma cells in vitro.
They tested approximately 750,000 small molecules from the ChemBridge library to identify drug candidates with specific affinities for partially unfolded GFAP. One, named MSR1, was particularly effective in reducing protein aggregation and pathology in various AD models (human cells or C. elegans) and could serve as a lead compound in further searches.
All the research methods chosen by the authors do not raise any major doubts. And the conclusions drawn from the results allow us to be more optimistic about future treatments for neurodegenerative diseases.
The research work presented in the above manuscript is very valuable from the point of view of the treatment of dementia-related diseases, therefore I believe that it fully deserves to be published in this journal.
Author Response
Thank you for your positive comments!
Reviewer 3 Report
The manuscript by Ganne and colleagues entitled “Glial Fibrillary Acidic Protein: A biomarker and drug target for Alzheimer’s disease” describes the use of GFAP, highly overexpressed and differentially phosphorylated in AD hippocampus, as a valuable biomarker in Alzheimer disease and/or other aggregation-associated diseases. Manuscript is remarkable and its interest, in terms of novelty and relevance, is high. The experimental design is correct and the results are clear-cut. Therefore, this paper is suitable for publication at this stage in the Journal. However, I have some (minor) comments:
1. What is the advantage of the glial fibrillary acidic protein (GFAP) respect to other significant proteins or receptors, such as iba-1, toll-like receptor 4 (TLR-4) which are increase in neurodegenerative disorders?
2. Grammatical and formatting errors are present within the text, please check it.
3. It is better to summarize the results of experimental studies using tables (including biomarker, model, effect and mechanism, etc), which will be more direct and clear.
Author Response
Thank you for your comments. Our replies are given below.
- What is the advantage of the glial fibrillary acidic protein (GFAP) respect to other significant proteins or receptors, such as iba-1, toll-like receptor 4 (TLR-4) which are increase in neurodegenerative disorders? We have published numerous papers on other proteins that increase in neurodegenerative aggregates. The most important common feature is that several such aggregate proteins appear to be hyperphosphorylated in disease-associated aggregates. Our surmise is that the deciding factors are kinase upregulations that may have diverse targets, each predisposed by the PTM to aggregation. This is the rationale for our pursuit of upstream kinases, and we will try to emphasize that in the revised text.
- Grammatical and formatting errors are present within the text, please check it. We will check again for grammar and formatting errors.
- It is better to summarize the results of experimental studies using tables (including biomarker, model, effect and mechanism, etc), which will be more direct and clear. There is one supplemental table, which we agree is a useful mode of data presentation.